# Assessing Patient Experience and Attitude: BSC-PATIENT Development, Translation, and Psychometric Evaluation—A Cross-Sectional Study

**DOI:** 10.3390/ijerph19127149

**Published:** 2022-06-10

**Authors:** Faten Amer, Sahar Hammoud, David Onchonga, Abdulsalam Alkaiyat, Abdulnaser Nour, Dóra Endrei, Imre Boncz

**Affiliations:** 1Doctoral School of Health Sciences, Faculty of Health Sciences, University of Pécs, H-7621 Pécs, Hungary; hammoud.sahar@etk.pte.hu (S.H.); onchonga.david@etk.pte.hu (D.O.); 2Institute for Health Insurance, Faculty of Health Sciences, University of Pécs, H-7621 Pécs, Hungary; endrei.dora@pte.hu (D.E.); imre.boncz@etk.pte.hu (I.B.); 3Division of Public Health, Faculty of Medicine and Health Sciences, An Najah National University, Nablus P.O. Box 7, Palestine; abdulsalam.alkaiyat@unibas.ch; 4Faculty of Economics and Social Sciences, An Najah National University, Nablus P.O. Box 7, Palestine; a.nour@najah.edu; 5National Laboratory for Human Reproduction, University of Pécs, H-7621 Pécs, Hungary

**Keywords:** balanced scorecard, patient engagement, satisfaction, hospital, performance evaluation, quality

## Abstract

Health care organizations (HCO) did not consider engaging patients in balanced scorecard (BSC) implementations to evaluate their performance. This paper aims to develop an instrument to engage patients in assessing BSC perspectives (BSC-PATIENT) and customize it for Palestinian hospitals. Two panels of experts participated in the item generation of BSC-PATIENT. Translation was performed based on guidelines. Pretesting was performed for 30 patients at one hospital. Then, 1000 patients were recruited at 14 hospitals between January and October 2021. Construct validity was tested through exploratory factor analysis (EFA) and confirmatory factor analysis (CFA). Additionally, the composite reliability (CR), interitem correlation (IIC), and corrected item total correlation (CITC) were assessed to find redundant and low correlated items. As a result, the scales had a highly adequate model fit in the EFA and CFA. The final best fit model in CFA comprised ten constructs with 36 items. In conclusion, BSC-PATIENT is the first self-administered questionnaire specifically developed to engage patients in BSC and will allow future researchers to evaluate the impact of patient experience on attitudes toward BSC perspectives, as well as to compare the differences based on patient and hospital characteristics.

## 1. Introduction

### 1.1. Health Care System in Palestine

The performance of health care services is adversely affected by long waiting times, inefficiency, low productivity, burnt-out medical staff, and dissatisfied patients [1]. In addition to these universal challenges, the health care system in Palestinian territories has also been slapped by political and economic conflicts. Therefore, it is described to be incoherent and inadequate [2,3]. The 87 hospitals in Palestinian territories have five major types based on administrative type: 28 public, 39 nongovernmental organizations (NGOs), 17 private, two military, and one United Nations Relief and Works Agency for Palestine Refugees in the Near East (UNRWA) [4]. Military hospitals are not yet operating in West Bank. The bed percentage per administrative type is approximately 59% public, 26% NGO, 14% private, and 1% UNRWA [5]. These hospitals are distributed as seven in eastern Jerusalem, 53 in West Bank, and 30 in Gaza [6]. The geographic separation with the disrupted mobility between these territories, added to the blockade of the Gaza strip, the checkpoints in West Bank and Jerusalem, the separate de facto government health systems in Gaza and West Bank, the heavy reliance on external health financing, and the dependence on direct household expenditures imposed further challenges on improving the Palestinian health care system [2,7,8,9]. The spread of Corona virus-19 (COVID-19) has added an additional challenge. A recent study [10] referred to the COVID-19 era in conjunction with political conflict to have a double epidemic effect on Palestinian territories, which eventually impacted the Palestinian health system and health care organizations (HCO) performance during the pandemic.

### 1.2. History of Balanced Scorecard (BSC)

In 1992, Norton and Kaplan proposed the initial design of the balanced scorecard (BSC), which incorporated four perspectives: financial, customer, internal process, and knowledge and growth [11]. In some previous implementations of the BSC, the last perspective was also termed the learning, innovation, technology and development perspective [12].

The first generation of BSCs contained only the four perspectives steered by the organizational strategy. Figure 1 depicts the first generation of BSCs. In the second generation, researchers demonstrated the existence of causal links between the key performance indicators (KPIs) of these four perspectives [13]. These connections were referred to as BSC’s strategic map. See Figure 2. The third generation, which incorporated objectives and action plans for each KPI, was then introduced.

The environmental and social perspective of sustainability was later added as the fifth pillar of BSC [14]. However, our recent systematic review of BSC implementations in the health care sector revealed that the management perspective should also be incorporated in BSC design. In this review, the 797 KPIs were reduced into 45 subdimensions after classification and regrouping. The reassembly of these subdimensions yielded 13 major dimensions:
Figure 2Duke University Health System Strategic Map [15].
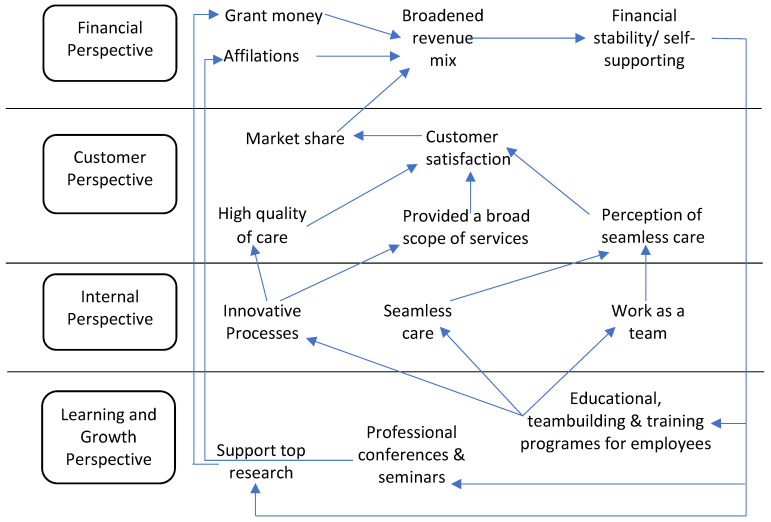


Financial, efficiency and effectiveness, availability and quality of supplies and services, managerial tasks, health care workers’ (HCWs) scientific development being error-free and safe, time, HCW-centeredness, patient-centeredness, technology, and information systems, community care and reputation, HCO building, and communication. See Figure 3.

We summarize the perspectives, major dimensions, and subdimensions that were more frequently used and deemed essential by health care managers worldwide.

### 1.3. The Impact of BSC

Two reviews focused on studying the effect of BSC, one of which analyzed the impact qualitatively [16] and the other presented a few instances of the positive influence [17]. This showed that no complete or rigorous scientific methodology has been reported until 2022 to evaluate the effect of BSC adoption in HCO. Given the lack of research on this topic, we performed a systematic review in which we assessed the impact of implementing the BSC on three attributes that represent the latest affected perspectives in the strategic maps [17,18]: HCWs’ satisfaction, patient satisfaction, and financial performance. As a result, BSC implementation proved to positively improve the financial performance of HCOs [19]. Furthermore, we found that BSC was beneficial in enhancing the patient satisfaction rate. Additionally, BSC influenced the health care workers’ (HCW) satisfaction rate, but to a lesser extent [19]. Despite the fact that BSC has a beneficial influence on patient satisfaction, prior implementations of BSC have solely focused on measuring patient satisfaction. One implementation at HCO in Afghanistan [20] created the community scorecard (CSC) to include the community in the assessment of the BSC. However, none of the studies included patients in the process of evaluating BSC [12,19,21]. Involvement of patients in this process could result in even higher levels of patient satisfaction. In addition, it will assist HCO managers and researchers in better understanding the BSC strategic maps as well as the causal links between KPIs based on the perceptions of patients.

In contrast to other performance evaluation (PE) tools, which primarily focus on analyzing the internal perspective, the BSC is regarded as a comprehensive approach for PE, as it involves the analysis of six perspectives [12]. For that, BSC implementations utilized different sources to conduct the PE of HCOs [12,19], including hospital records, patient satisfaction questionnaires, patient and HCW interviews, and observations. Additionally, BSC reviews [12,19] showed that only a few BSC implementations utilized validated scales to evaluate patient satisfaction, such as the Press Ganey questionnaires [22,23]. The patient satisfaction perspective is important since patients represent the hospitals’ end receivers of health care services. However, researchers have pointed to the importance of the engagement of patients (EoP) in the process of health policy planning, evaluation, and delivery improvement [24,25]. Additionally, patient feedback was proven to positively impact performance in HCO [14]. Strategies to support EoPs include patient needs assessment, communication skills improvement, managing patient conflicts and complaints, maintaining patient confidentiality, patient training, and asking patients to review outputs by assessing their perceptions and experiences [25,26]. It is not sufficient to perform the PE of HCO based on manager and hospital records only; a focus on EoP among the selection of the KPIs at HCO was recommended [24]. However, BSC reviews referred to the lack of patient and family member involvement in the evaluation process of BSC [12,19,21,27].

The first aim of this research was to develop a comprehensive instrument (BSC-PATIENT) that is able to assess: 1. patient experiences in light of BSC perspectives, 2. patient PI regarding BSC perspectives, and 3. patient satisfaction and loyalty attitude. The second aim of this research was to customize the developed instrument at Palestinian hospitals, translate it into Arabic, and validate it.

## 2. The Conceptual Framework

In our conceptual model we considered the impact of BSC six perspectives which resulted at our previous systematic review and their underlying dimensions [12]. We also built it based on the psychological definitions of experiences and attitudes [28,29] and the previous literature regarding patient attitudes [28,30,31,32,33]. Experiences and perceptions enable people to act in a particular behavior and develop an image, satisfaction, or loyalty attitude [29]. Figure 4 represents our conceptual model.

### 2.1. The Experience

Experience is defined as an event that was lived through [29]. Patient experiences at HCO are formed upon receiving the health care service or treatment. Becoming aware of the events, objects, or relationships utilizing senses or observation results in experience perceptions [29].

### 2.2. Attitudes

Attitudes form directly as a result of experiences. There are three types of attitudes, which are sometimes referred to as ABCs of attitude. First, the affective component is how the object, person, issue, or event makes someone feel. The behavioral component is how attitude influences someone’s behavior. The cognitive component is someone’s thoughts and beliefs about the subject. An example of attitude is image perception, satisfaction, and loyalty. Such evaluations are often positive or negative, but they can sometimes also be uncertain [28].

#### 2.2.1. Patient Satisfaction Attitude

Satisfaction is the most commonly used metric by managers to assess customer perceptions [30]. Satisfaction does not always lead to loyalty. However, loyalty often begins with a sense of satisfaction [31]. Studies have found that patient satisfaction either plays a direct impact on loyalty attitudes or acts as a moderating variable between service quality and loyalty attitudes [32].

#### 2.2.2. Brand Preference Attitude

Brand preference is the degree to which consumers prefer a specific brand relative to competing alternatives. It is considered an essential component of customer loyalty [30].

#### 2.2.3. Perceived Quality (PQ) Attitude

Studies have proven that PQ exerts an indirect influence on patient loyalty. A rival hypothesis referred to satisfaction as a mediator between PQ and loyalty [32].

#### 2.2.4. Perceived Image (PI) Attitude

A hospital PI was defined as the sum of beliefs, ideas, and impressions that a patient holds toward a particular hospital [34]. Patients usually form a PI of a hospital from their own past treatment experiences relative to the PIs of competing hospitals [33]. A positive PI of a bank was found to significantly improve the PQ. Therefore, in health care, a positive hospital PI may positively influence PQ. However, a recent review showed that this has not yet been studied [33].

#### 2.2.5. Loyalty Attitude

A loyalty attitude is a behavioral intention that reflects faithfulness and allegiance to something [29]. In the marketing management field, Kotler and Keller (2015) defined loyalty as a deeply held commitment to rebuy or repatronize a preferred product or service in the future, despite influences to cause switching behavior [35]. A study revealed a need to use multiple indicators to predict customer loyalty behavior, such as customer satisfaction, brand preference against competitors, intention to return or repurchase, and willingness to recommend [30]. Moreover, customer behavior trends in the past were a good predictor of future customer behavior. It is important to emphasize that loyalty refers to customers’ actual conduct, regardless of their attitudes or preferences. However, assessing customer loyalty attitudes can help predict their loyalty behavior in the future [36].

##### Repurchase Intention Attitude

Researchers have used repurchase intentions to help predict future purchasing behavioral intentions and loyalty [30]. On the other hand, customer retention behavior is defined as customers stating the actual continuation of a relationship with the organization. It is well known in marketing that past customer behavior tends to be a relatively good predictor of future customer behavior. However, most researchers focus on assessing repurchase intention attitudes and neglect assessing actual customer retention behavior [30].

##### Willingness to Recommend an Attitude

Word-of-mouth intention has been of importance to researchers in the past 30 years. Thus far, there is very little scientific research relating the intention of the recommendation to the actual recommendations [30].

## 3. Methods

### 3.1. Research Design

This is part of a broad project that aims to strategically develop Palestinian hospitals using BSC. This research is a cross-sectional study. The questionnaire was created and validated based on the key authors Kaplan and Norton’s theortical framework [11] and the best practices for developing and validating health and behavioral scales [37].

### 3.2. Item Generation

The first panel consists of five authors in this research. Two researchers in health management (first and fourth), two hospital managers who are also expert researchers in health management (sixth and seventh), and one expert in the BSC tool (fifth) provided expert input on all stages of instrument development. First, we performed a systematic review [12], in which 797 KPIs were extracted from 36 BSC implementations at HCO worldwide. Then, categorization and regrouping of these KPIs resulted in 45 subdimensions and 13 major dimensions that are frequently used by health care managers and are important for PE and the strategic development of HCO [12]. Next, this panel performed a four-round Delphi method [38]. In the first round, the panel prepared a survey for hospitals’ top managers to rate the resulting 45 subdimensions on a 10-point semantic scale, based on their importance for the strategic development of their hospitals. A description for each subdimension using the shortlisted KPIs was included in the manager survey. In the second round, the panelists reviewed the item face validity per subdimension [39]. Next, the first author asked a second panel consisting of 13 top hospital managers from 4 Palestinian hospitals to answer this survey individually. Additionally, hospital managers were asked to mention whether they considered any other subdimension or KPI that was not listed as essential. The subdimensions with an average score above 0.7 were chosen for the next step based on their ratings. In the third round, the first panel reviewed the resulting important subdimensions at the previous step and decided which subdimensions the patients could be engaged in their evaluation. As a result, 24 subdimensions resulted. In the fourth round, the panelists revised each item wording and clarity to patients. As a result, 52 items remained. In the fourth round, the panelists rated the relevance and importance for each remaining item based on four- and three-point ordinal scales, respectively [40]. Next, the first author calculated the content validity ratio (CVR), the item content validity index (I-CVI), the scale content validity index (S-CVI), and universal agreement among experts for the content validity index (CVI-UA) to assess the content validity per item and scale [40]. Only the items rated 0.99 or above in CVR were included as per Lawshe guidelines [41]. However, dimensions that scored 0.80–0.99 indicated the need to be revised. For the CVI, items that scored less than 0.60 were eliminated. Items that scored 0.6–0.79 were revised [40]. See Figure 5.

The panelists suggested using a three-point Likert scale: yes, neutral (I do not know), and no. This choice was due to the high number of the remaining items, the evidence of a high nonresponse rate of patients to the five-point Likert scale-validated tools [42,43,44,45], and the possibility for assessing item availability using yes/no questions. Additionally, this was found to lead to a faster and better item response, specifically considering the pandemic load on hospitals. All authors were asked to revise the instrument, and the final modifications were made accordingly.

### 3.3. Linguistic Validation and Translation

BSC KPI, balanced scorecard key performance indicators; CVI, content validity index; CVR, content validity ratio; CR, composite reliability; IIC, interitem correlation; CITC, corrected item-total correlation.

Since the dimensions resulting from the systematic review were in English, the questionnaire items were initially developed in English. Then, they were translated to Arabic. All translations were prepared as per the translation and validation guidelines [46]. The first author performed a final review to produce the final corrected translation. An expert checked the final form in the BSC, and minor modifications were recommended.

### 3.4. Pretest and Internal Consistency

The first version of the questionnaire was piloted in one NGO hospital in the south of West Bank. For that, 30 patients were asked to answer the first version of the questionnaire. They were asked to write their comments regarding language simplicity. The time needed to complete the questionnaire was also recorded. Items were coded before performing the analysis by IBM SPSS statistics 21 software. Then, Cronbach’s alpha was calculated for each perspective to evaluate the internal consistency [47], and values above 0.6 were considered acceptable. Based on the results, some items were modified or deleted.

### 3.5. Sampling Procedure and Power Calculation

Institutional Review Board (IRB) approval for this research was received on 31 May 2020. All methods described in this study were approved by the Research and Ethics Committee at the Faculty of Medicine and Health Sciences at An Najah National University with the reference code number (Mas, May/20/16). Afterward, requests at 15 hospitals in West Bank and three hospitals in Jerusalem were applied between June and December 2020. The hospitals were selected using a convenience sample. However, the total number of beds per administrative type and governorate was considered for choosing the participants (HCO and patients). Public hospital approval was first applied to the Palestinian Ministry of Health. Then, the request was applied to each hospital individually for all hospital types. The final form of the questionnaire was distributed between January and October 2021. The sample size was calculated according to the Steven K. Thompson sample size equation [48]:
$$n=\frac{N\times p\left(1-p\right)}{\left[N-1\times \left(d^{2\;}÷z^{2}\right)\right]+p\left(1-p\right)}$$

where *n* is the sample size, *N* is the population size, *p* is the estimated variability in the population (0.5), *d* is the margin of error (0.05), and the *z* score is at the 95% confidence interval (1.96). In our study, *N* was the population volume in the Palestinian territories [4]. Therefore, the needed sample size was found to be *n* = 385 patients. Additionally, studies considered 300 participants as a good sample size to successfully run each exploratory factor analysis (EFA) and confirmatory factor analysis (CFA) or 5 respondents per parameter [49,50,51]. Splitting the sample to perform EFA and CFA is recommended to perform construct validity [52]. Therefore, a total of 1000 questionnaires were distributed, anticipating a lower response rate during the pandemic.

### 3.6. Data Collection and Participants

The first author and four medical students at An-Najah University collected the data. Each medical student received three hours of training on BSC and the data collection steps and ethics by the first author. Tasks and hospitals were delegated to them according to their living area: eastern Jerusalem and north, middle, and south of West Bank. The Gaza Strip was excluded due to the political situation and accessibility obstacles during the study. Moreover, five hospitals were excluded: two military hospitals that were not operating yet, one psychiatric hospital, and two rehabilitation hospitals. We sought variation in our sample regarding hospital size, area, and administrative type. For that, the maximum variation sampling strategy was used. The number of hospitals and the number of beds per administrative type were considered upon recruiting the sample [4]. The patients were conveniently chosen based on their willingness to participate in this research.

Printed questionnaires were distributed to respondents instead of sending the questionnaires via email to reduce nonresponse bias [53]. Additionally, all participants were asked to agree on participation in a consent form that is coherent with the Declaration of Helsinki ethical principles [54]. Patients were informed that participation was confidential. Additionally, all patients were informed that participation was voluntary, so they could refuse participation in the study or withdraw at any time. To reduce the response bias [53], the “I don’t know (neutral)” answer was added as an option, since experiences and attitudes can sometimes be uncertain [28]. Second, the data collectors ensured that the number of missing answers was minimized by checking the questionnaires upon retrieval. In case of missing parts, they drew the participant’s attention to answer them. When entering data, if any questions were found to be still missing, they were entered as I don’t know.

The inclusion and exclusion criteria were set to be a Palestinian patient above 15 years old of any gender. Outpatients should have finished receiving medical care at the assessed hospital or had received medical care at least once previously and returned to the same hospital. Inpatients should have been admitted for at least one day. The following departments were included: emergency room, internal medicine, surgery, gynecology, and pediatrics. In the emergency department, the questionnaires were completed by the patient companions. Additionally, in the pediatric department, the questionnaires were completed by one parent of the child. For the rest, questionnaires were completed by patients themselves; unless they were unable to complete the questionnaire, the questionnaires were read to them by the data collector or a family member and completed according to patient answers. To distinguish, a question was added to ask the respondent if his responses were based on his own, family, or friends’ experiences.

### 3.7. Statistical Analysis

Normality was tested using the Shapiro–Wilk test. The frequencies were used to analyze patient sociodemographics and the participating HCO characteristics. Our sample was split based on admission status to assess construct validity using EFA and CFA. EFA was performed for the inpatient sample using principal axis factoring with the Promax rotation method [55] in IBM SPSS statistics 21 software. The Kaiser–Meyer–Olkin (KMO) and Bartlett’s sphericity tests were tested to determine the adequacy of the EFA [56]. The inclusion or exclusion of a component was determined by an eigenvalue ≥1 [57] and the visual assessment of Cattell’s scree plot [58]. Item inclusion or exclusion was determined by a factor loading ≥0.50 and factor loadings on the assigned construct higher than all cross-loading of other constructs [50].

Second, CFA was performed for the components that resulted in EFA using the outpatient sample. The maximum likelihood estimation method in IBM Amos 23 Graphics software (IBM, Wexford, PA, USA) was applied. The goodness of fit for the competing models was evaluated through the most commonly used fit indices. Minimum discrepancies were divided by degrees of freedom less than five and closer to zero, *p* value higher than 0.05, goodness-of-fit index (GFI), comparative fit index (CFI), Tucker–Lewis’s index (TLI), and cutoff values close to 0.95. Additionally, a root mean square error of approximation (RMSEA) <0.06 and standardized root mean square residual (SRMR) value <0.08 are needed before we can conclude that there is a relatively good fit between the hypothesized model and the observed data [59,60]. Item inclusion or exclusion in CFA was determined by a factor loading ≥0.50.

Third, the interitem correlation (IIC) and the corrected item-total correlation (CITC) were calculated [61]. In this study, items with a correlation higher than 0.9 were considered redundant and deleted [62]. A correlation of 0.3 was considered the lower limit. Additionally, the composite reliability (CR) per construct was evaluated after performing CFA. CR is preferred over Cronbach’s alpha, specifically in structural equation modeling [63]. In the current study, a CR ≥ 0.6 was considered sufficient [64,65].

Finally, the Fornell-Lacker criterion was used to evaluate convergent and discriminant/divergent validities [66]. The average variance extracted (AVE) was considered adequate for convergent validity if it was higher than 0.5. However, if a value <0.5 with CR > 0.6, the convergent validity of the construct was still considered adequate [66]. To establish discriminant validity, the square root of the AVE (SQRT) should have a greater value than the correlations with other latent constructs [64]. Additionally, construct uniqueness was evaluated depending on the value of Spearman correlation (r) with other constructs at the same scale. Researchers recommended the separation of dependent and independent variables since the correlation between them can be misleading in assessing discriminant validity [67]. Therefore, we assessed r for the independent and dependent constructs separately. Then, r was described as negligible when r < 0.2, low (r = 0.2–0.49), moderate (r = 0.5–0.69), high (r = 0.7–0.85), or very high (r = 0.86–1.00). In this study, the absence of high or very high r between the subscale constructs indicated discriminant validity [68].

## 4. Results

### 4.1. Item Generation and Scoring

The demographics and characteristics of the second-panel hospital managers are shown in Table 1. The content validity resulted in removing one item and indicated that a revision is needed for eight items. The revised items required either further clarification and rewording or modification for specific participants. For example, the CVR results indicated that financial and price items should not be included for nonprofit hospitals. Additionally, the CVI results showed that particular items were relevant only to inpatients. This step raised the S-CVI, CVI-UA, and CVR from 0.90, 0.63, and 0.95 to 0.95, 0.78, and 0.97, respectively.

### 4.2. The Instrument’s Structure and Items

The patient sociodemographics and hospital characteristics section included age, gender, scientific degree, working sector, insurance availability, and type. Moreover, the number of visits to the evaluated hospital compares the attitudes of the new and previous customers. The number of earlier visits is considered necessary in the analysis since past customer behavior tends to be a good predictor of future behavior [19]. Moreover, the information source on which the respondent evaluation was built was recorded since perceptions and attitudes may emerge from direct personal experience or from observing other people’s experiences, such as family and friends’ experiences [20]. The second section of the questionnaire was designed to measure patient experiences in light of BSC perspectives and their attitudes toward them, including patient satisfaction, PQ, PI, and loyalty.

#### 4.2.1. The Financial Perspective

It evaluated the health services and medication’s price affordability. This section was answered only by patients who did not have insurance.

#### 4.2.2. The Internal Perspective

This perspective assessed safety, time, and service availability. On the other hand, the PI of the cure rate, accuracy, complications, and PQ of services and medication were measured in the attitude section.

#### 4.2.3. The Knowledge and growth Perspective

Information and training provided to patients were assessed in the experience section. Additionally, we assessed the PI of hospital technology and employee competencies in the attitude section.

#### 4.2.4. The Customer Perspective

It assessed patient-centeredness and the HCW–patient communication experience. The attitude section assessed actual patient satisfaction and loyalty attitudes. In previous studies, validated items for loyalty measurement included satisfaction measurement and loyalty attitude measurement, specifically the recommendation and return intentions [30,33]. Using a single item to directly assess actual patient satisfaction was suggested to be better than its assessment through multidimensional items [69].

#### 4.2.5. The Environmental Perspective

It evaluated the hospital building environment and hospital capacity, ease of access, and female concern experiences. On the other hand, a comparison with the other hospitals’ medical and social PIs was included in the attitude section.

Finally, four items were reversed in the instrument, PIN9, which assessed the long waiting time. Additionally, PIN4, PIN5, and PIN6 assessed readmission, referral to other hospitals, and postoperative infection probability expectations, respectively.

#### 4.2.6. The Managerial Perspective

As there is no direct contact experience between patients and hospitals’ managers, we evaluated the hospital administrative type and the accreditation status in this perspective. So, we can study the impact of these factors on patient attitudes.

### 4.3. The Pretest and the Internal Consistency

The pretest was performed at one NGO hospital in the south of West Bank. Patients found the length of the questionnaire appropriate. Additionally, the layout was well accepted and clear. They gave specific minor comments that were incorporated. These corresponded to the rewording of a few items. The time for completing the questionnaire was less than 10 min.

Consequently, few modifications were made after piloting. Cronbach’s alpha was calculated per BSC perspective. All perspectives had a Cronbach’s alpha above 0.7 at the pretest, except for the environmental perspective, which was 0.59. Hence, some of its items were moved to other perspectives, and five items were deleted. As a result, 52 and 50 items remained for inpatients and outpatients, respectively.

### 4.4. Linguistic Validation and Translation

The final English and Arabic questionnaire forms were ready for use.

### 4.5. Sample Size and Characteristics

Since the research coincided during the COVID-19 pandemic, hospital approvals took six to nine months until they were received. Only 15 hospitals out of 18 agreed to participate. The UNRWA, The United Nations Relief and Works Agency for Palestine Refugees in the Near East; NGO, Non-Governmental Organization.

Data collection was performed between January and September 2021. The data from the pretest at one hospital were excluded. Next, we distributed 1000 questionnaires at the remaining 14 hospitals. As a result, 740 were returned (response rate was 74%). The characteristics and sociodemographics of the respondents are shown in Table 2 and Table 3.

### 4.6. Statistical Analysis

The statistical analysis using the Shapiro–Wilk test showed that the data were not normally distributed, so nonparametric tests were used. Then, construct validation was assessed for the instrument.

#### 4.6.1. Construct Validity in EFA

EFA resulted in 46 items with loadings higher than 0.50 for 16 components. Eigenvalues for all components were higher than one. The KMO was 0.813, reflecting a very high sampling adequacy [56,64], and Bartlett’s test was also significant. The cumulative variance was 67.414%. See Table 4. The 12 components were patient attitude toward BSC perspectives (BSCP ATT), patient experience (PT EXR), service experience (SERV EXR), price experience (PR EXR), building experience (BUIL EXR), access experience (ACC EXR), complication perceived image (COMP IMAGE), technology experience (TECH EXR), information experience (INFO EXR), hospital social responsibility perceived image (HSRP IMAGE), and waiting time experience (WT EXR). One item (SAT2) loaded on the 12th component.

However, this item had a higher loading on the BSCP ATT. None of the specific inpatient items had loadings higher than 0.50. Moreover, the scree plot showed the necessity of deleting the last three components.

#### 4.6.2. Construct Validity in CFA

The resulting nine components in EFA were tested in the Amos program. The model was edited based on the item loadings, model fit indices, and calculations in the convergent, discriminant, CR, IIC, and CITC at the next step until we arrived at the best model. First, adding two items that did not have loadings to the INFO EXR construct showed good loadings in CFA. The same was true for the BSCP ATT and TECH IMAGE constructs. Second, splitting the BUIL EXR component into two separate constructs, building environment experience (BUILENV EXR), and building capacity experience (BUILCAP EXR), improved the item loadings and the model fit. Third, PEN9 and PLE7 items were removed from the PT EXR construct because they have loadings lower than 0.50. On the other hand, PIN 14 and PIN 16 were added to BSCP ATT construct since both had loadings higher than 0.50 and improved the model fit. Moreover, merging the TECH IMAGE and COMP IMAGE items at the BSCP ATT construct resulted in loadings lower than 0.5 and IIC lower than 0.30. Hence, three separate constructs in the attitude section were decided. Finally, the modification indices in the Amos program were utilized to improve the model. The final model revealed that the CMIN/df, CFI, GFI, TLI, RMSEA, and SRMR indices in CFA were above or close to the cutoff points, reflecting a good fit model. Nevertheless, the *p* value was <0.001, which can be referred to as its sensitivity to normality. See Figure 6 and Table 5. To see the items which did not load in EFA, the items which were tested in CFA, and the final resulted items, refer to the Appendix A.

#### 4.6.3. Composite Reliability and Interitem Correlations

The composite reliabilities for all constructs were higher than 0.6 except the SERV EXR construct. However, this construct’s IIC and CTIC were higher than 0.3. The other constructs also had IICs higher than 0.3, and their CITC ranged from 0.328–0.853, reflecting satisfactory IIC and CITC. See Table 6.

#### 4.6.4. Convergent and Discriminant Validity

Convergent validity was less than 0.5 for BSCP ATT, BUILENV EXR, PTCOMINF EXR, SERV EXR, and COMP_IMAGE. However, the CR, IIC, and CITC showed satisfactory results [66], except for the SERV EXR, which had a CR equal to 0.50 but an IIC and CITC higher than 0.3. On the other hand, the square roots of the AVE were higher than the off-diagonal correlations between constructs. Additionally, a lower correlation between constructs indicates each construct’s uniqueness. The correlations between the independent constructs were either negligible or low, except between two constructs, the PT EXR and INFO EXR, which were moderate. Merging the two constructs lowered the loadings and the model fit indices in CFA. The same was perceived regarding merging the BUILENV EXR and BUILCAP EXR constructs. Consequently, separate constructs were determined, as mentioned earlier. In regard to the independent constructs, negligible or low correlations existed among them. Neither high nor very high correlations existed between the independent constructs. Therefore, this establishes discriminant validity and the uniqueness of the independent constructs. The same holds true for the dependent constructs. In other words, convergent validity was met for all constructs except SERV EXR. In comparison, discriminant validity was met for all constructs, as shown in Table 7 and Table 8.

## 5. Discussion

### 5.1. Discussion of the Main Results

In agreement with this paper’s aim, it was possible to build a valid and reliable instrument. BSC-PATIENT is the first validated instrument to engage patients in the evaluation of hospitals by measuring their experiences and attitudes toward the hospital based on the BSC perspectives: the financial, internal, knowledge and growth, customer, and environmental perspectives. The deployment of this instrument at BSC implementations and PEs in general will improve patient satisfaction and allow a better understanding of BSC strategic maps based on patients’ experiences and attitudes.

Our findings showed that patient attitude toward all BSC perspectives and dimensions loaded on one construct, except the images of technology and complications, loaded separately.

The instrument was customized to be compatible with Palestinian hospitals. Statistics revealed that out-of-pocket household payments constituted 39.8% of the Palestinian territories’ total health care expenditures in 2018 [71]. This number is close to the results in our sample, which showed that 14.73% of patients did not have any insurance, and 19.32% had private insurance. Additionally, our analysis shows that another 35.41% or 1.49% of our sample had public or UNRWA insurance, respectively, but were receiving treatment at an NGO or private hospital at the time of the study. This situation indicates that the patients either made out-of-pocket payments or that the government paid a medical referral to private or NGO hospitals [4]. Therefore, incorporating the financial perspective consideration in this paper proved to be vital. Additionally, many BSC implementations in Afghanistan and Bangladesh revealed the need to consider the social and cultural perspective in evaluation, specifically female attentiveness concerns [20,72,73,74,75]. The authors believed that this was also the case in Palestine, so the BSC-PATIENT included such items. However, in different cultures, this may not be important. Hence, these items can be removed or replaced with other customized environment-related items. Finally, the technology perspective varies among Palestinian hospitals. Even though the Ministry of Health Hospitals and many other private hospitals have adopted the health medical information system for years, some hospitals still use the manual system for documentation. The authors also considered this perspective important in this evaluation.

The causal relationships between BSC dimensions that were described in BSC strategic maps may impose a challenge on producing a good fit model, specifically discriminant validity. Despite this challenge, our model proved satisfactory construct, convergent, and discriminant validity. The composite reliability was higher than 0.6 for all constructs except the SERV EXR construct. This may indicate that a separate evaluation for this construct item is needed. Moreover, the IIC and the CITC were satisfactory. In general, this questionnaire proved reliable and valid for engaging patients in hospital evaluations by measuring their experiences and attitudes toward Palestinian hospitals.

### 5.2. Comparison with BSC Implementations

The review of the dimensions utilized in BSC implementations [12] revealed that 77 percent of the implementations did not engage patients at any point in the assessment process. Instead, they relied only on hospital records and reports to evaluate the BSC perspectives. Patients were included in the remaining 22 percent of BSC implementations [72,74,75,76,77,78,79,80] to analyze only the patient satisfaction perspective. Although 11% of BSC implementations [20,74,80] included community members in the BSC perspective evaluation, none of the BSC implementations engaged patients in this process. In addition, patient interviews were utilized in each of the 22 percent of BSC deployments, but patient surveys were never used. This highlights both the significance of the BSC-PATIENT development and the originality of the study being conducted.

### 5.3. Comparison with Other Validated Instruments

#### 5.3.1. Service Quality Scale (SERVQUAL)

One of the most popular models to measure service quality is the 44-question SERVQUAL instrument [81]. However, SERVQUAL has been criticized for encountering various shortcomings [82,83]. First, numerous studies have questioned whether SERVQUAL is applicable as a generic scale for measuring service quality in all settings [82], as it was not initially designed for hospitals. In contrast, BSC-PATIENT was explicitly designed for hospitals. Second, the concept of “subtraction” in the SERVQUAL model is not equivalent to psychological function [82]. However, BSC-PATIENT was designed to be coherent with psychological definitions by distinguishing between experience observations and attitudes. Third, researchers uncovered some shortcomings of the discriminant validity at SERVQUAL [82]. They explained that reliability, responsiveness, assurance, and empathy dimensions were not distinct from each other and loaded into one factor in many studies due to the high degree of intercorrelation [82]. All BSC-PATIENT constructs passed discriminant validity. Fourth, SERVQUAL has been criticized for focusing on functional quality, not reputational quality [83]. This challenge was overcome in BSC-PATIENT through the separation of observations and attitudes.

#### 5.3.2. Press Ganey

Another commonly used instrument is Press Ganey [84], a 21-question instrument explicitly developed to measure hospital patient experience. However, Press Ganey also has a few shortcomings. Many studies using this instrument reported evidence of nonresponse bias [42,43]. The response rate for BSC-PATIENT was 75% despite the COVID-19 situation. Many patients commented that the questionnaire was interesting to complete. This can also be referred to as the simplicity of the three-point scale, unlike the five- and seven-point Likert scales, which can contribute to greater respondent burden and fatigue and may lead to higher refusal rates [69]. Finally, building, services, technology, price experiences assessing items, and patient attitudes were not considered necessary in Press Ganey.

#### 5.3.3. Hospital Consumer Assessment of Health Care Providers and Systems (HCAHPS)

The 29-question Hospital Consumer Assessment of Health Care Providers and Systems (HCAHPS) [85] is widely used in the United States of America (USA) to evaluate patient experiences. It incorporates eight dimensions. However, the response rate for this instrument was found to be low [44,45]. Additionally, accessibility, price, and technology experiences were neglected. Moreover, HCAHPS allows researchers to evaluate the overall patient satisfaction rate based on their subratings for different experience constructs, such as communication with HCW perception [44,45,86]. Although experience perceptions can predict patient attitudes, including satisfaction, a separate evaluation of experiences and satisfaction and a direct satisfaction assessment were recommended [69]. This point was taken into account when designing the BSC-PATIENT.

### 5.4. Strengths and Limitations

In general, this paper has several strengths. First, BSC-PATIENT is the first instrument that engages patients in BSC perspective assessment. Second, this instrument can determine patient attitudes, including PI toward BSC perspectives, PQ, and satisfaction and loyalty. Third, to our knowledge, this is the first paper to distinguish between patient experiences and patient attitudes, which will allow us to study the relationship between patient experiences and attitudes in future studies. Fourth, this instrument was customized to be used for all insurance, leadership, and admission statuses. Fifth, this instrument was designed based on KPIs extracted from BSC implementations in primary, secondary, and tertiary health care settings in low-, middle-, and high-income countries worldwide. Hence, the implementation of BSC-PATIENT can be generalized to different health care settings and countries. However, the instrument may need some customization based on the health care setting strategy and the country’s properties. For example, we customized the BSC-PATIENT at the environmental perspective based on Palestinian culture, the financial perspective based on administrative type, the knowledge and growth perspective based on the health information system in Palestine, and the few items specific for inpatients based on admission status. Finally, this paper offers a comprehensive hospital assessment from patient perspectives during COVID-19. To date, no study has assessed Palestinian hospital performance during this era. However, this instrument has some limitations. Despite this instrument assessing items such as patient education on infection control measures, it lacks COVID-19-specific items, as this instrument was designed before the COVID pandemic, so COVID-19-related items can be considered in future versions of the BSC-PATIENT instrument. Second, patient literacy was not assessed. However, the academic qualifications were evaluated at the demographics to be considered in the analysis. Third, measuring patient experiences in the past may involve a bias of recall. Additionally, participant bias may have occurred since the sample was convenient and the included hospitals agreed on participation. However, the high percentage of the included hospitals (30%) from the total number of hospitals at West Bank, and including all administrative type types from all regions, may have reduced the selection bias. Another limitation is that we could not validate this instrument in English due to our inaccessibility to English-speaking patients. Future research needs to consider testing the psychometric properties of BSC-PATIENT in an English-speaking country.

### 5.5. Practical Implications

Researchers and HCO managers are advised to utilize the BSC-PATIENT instrument in future BSC implementations. First, HCO managers will be able to highlight the strengths and weaknesses in BSC dimensions based on patients’ perspectives. Second, analysis of the BSC strategic maps based on patients will allow managers to highlight the predictors of patient satisfaction and loyalty. Third, HCO managers will be able to distinguish between the patients’ actual experiences and their attitudes. Analyzing the causal relationships between experiences and attitudes will provide insight for managers into which experiences should be improved to enhance patient attitudes. This will also guide managers in building their future action plans and how to allocate resources. Fourth, BSC-PATIENT can be utilized in the PE of HCO in general to evaluate a variety of dimensions instead of focusing only on patient satisfaction. The comprehensive analysis provided by this instrument will contribute to the health management field in general and will enhance patient satisfaction.

## 6. Conclusions

The BSC-PATIENT instrument was developed to engage patients in the PE of hospitals. This instrument was validated in Arabic and customized for Palestinian hospitals. This is the first instrument to engage patients in evaluating their experiences and attitudes toward the BSC perspective. It consists of 36 items; 21 items assessing patient experience observations and 15 items assessing patient attitudes. Both experiences and attitudes were designed based on BSC perspectives. The findings of this research showed adequacy in the psychometric properties of this instrument and suggest some recommendations for future research. First, we tested the psychometric properties of the BSC-PATIENT in English and other languages in different countries. Second, we consider developing instrumental BSC perspectives to engage other stakeholders in the PE of hospitals, such as doctors, nurses, and managers. Third, this instrument was used to assess the impact of patient experience on patient attitudes toward the hospital, specifically the PI, PQ, and satisfaction and loyalty. Fourth, managers must consider using a comprehensive approach for the PE of hospitals instead of limiting it to financial or internal indicators. Fifth, we compared the differences in patient experience and attitudes based on patient and hospital characteristics. Finally, enhancing patient engagement in the evaluation process instead of focusing on satisfaction alone must be considered in future BSC and PE implementations. Involving stakeholders in BSC’s comprehensive evaluation will lead to a better and deeper understanding of hospital PE.

## Figures and Tables

**Figure 1 ijerph-19-07149-f001:**
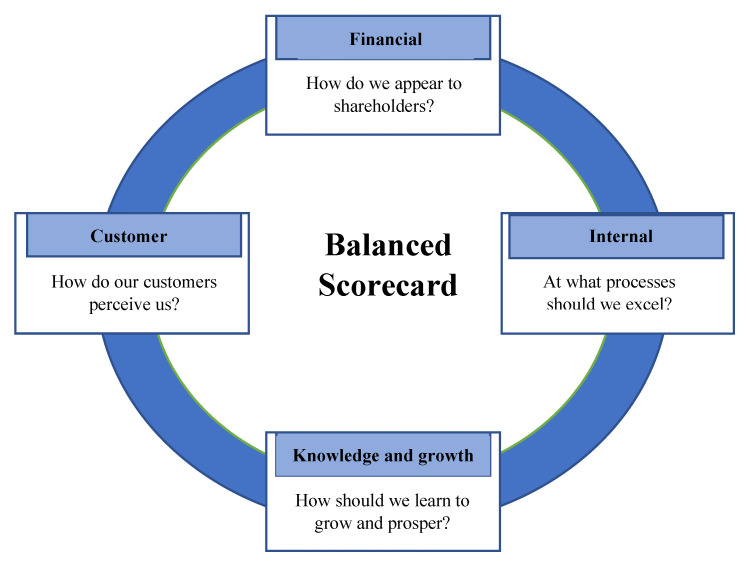
Balanced Scorecard perspectives [11].

**Figure 3 ijerph-19-07149-f003:**
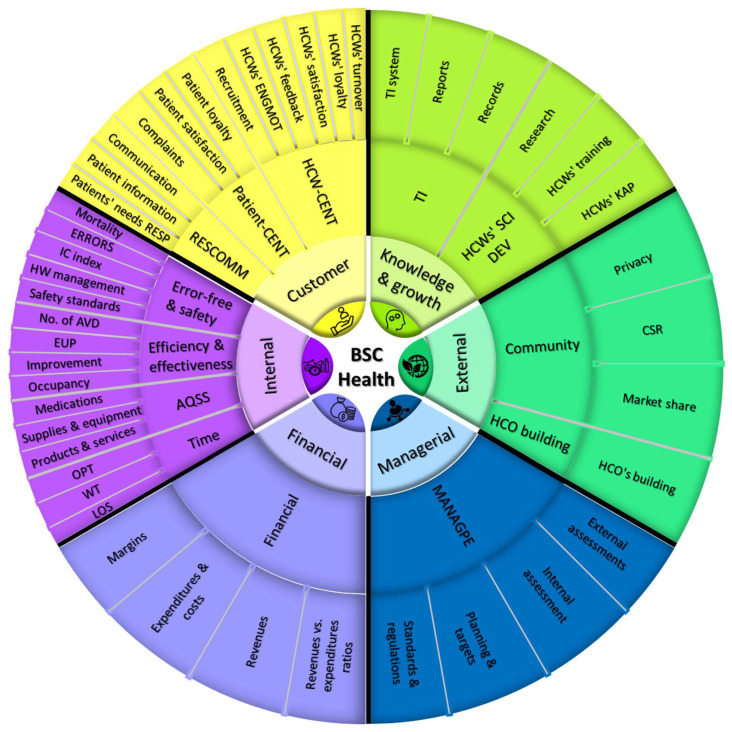
A summary of BSC perspectives in health care and their contents [12]. Figure legend: Summary of BSC perspectives and the underlying major and minor subdimensions for the PE of HCOs. Note: BSC, balanced scorecard; HCWs, health care workers; HCO, health care organization; IC, infection control; HW, health waste; WT, waiting time; LOS, length of stay; KAP knowledge, attitude, and practices; TI, technology and information; CSR, corporate social responsibility; ERRORS, errors, accidents, and complications; No. of AVD, number of admissions, visits, and diseases; EUP, efficiency, utilization, and productivity; AQSS, availability and quality of supplies and services; OPT, operation processing time; RESCOMM, response to patients’ needs; Patient-CENT, patient-centeredness; ENGMOT, HCWs’ engagement and motivation; HCW-CENT, HCW-centeredness; MANAGPE, managerial tasks and performance evaluation; SCIDEV, scientific development.

**Figure 4 ijerph-19-07149-f004:**
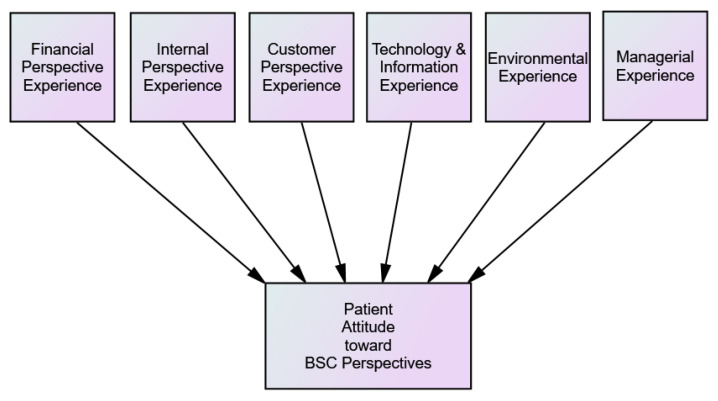
BSC-PATIENT conceptual model.

**Figure 5 ijerph-19-07149-f005:**
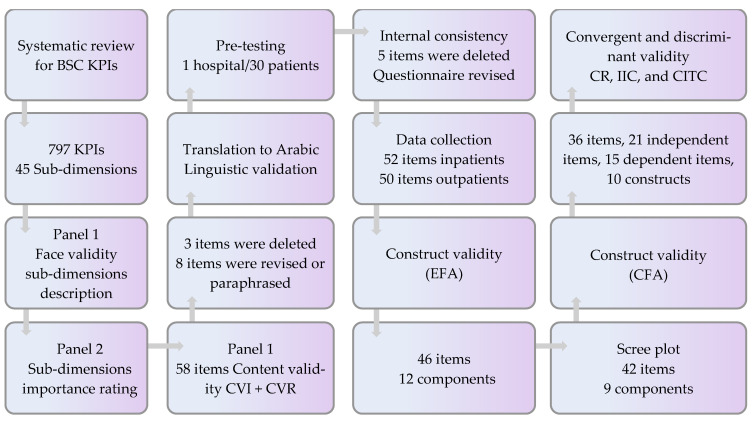
Flow chart for BSC-PATIENT development and psychometric validation.

**Figure 6 ijerph-19-07149-f006:**
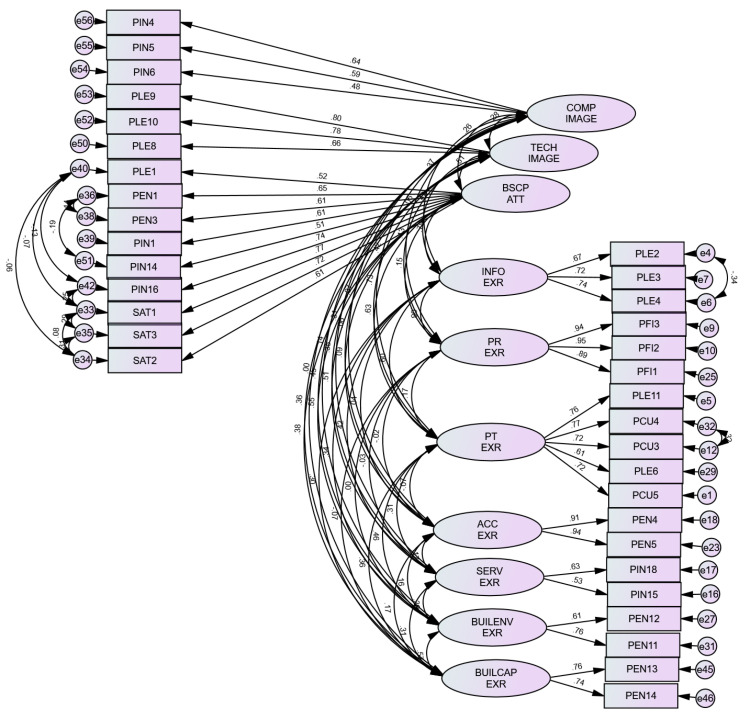
Confirmatory factor analysis (CFA). Independent items on the right side and dependent items on the left side. Note: COMP IMAGE, complications perceived image; TECH IMAGE, technology perceived image; BSCP ATT, patient attitude toward balanced scorecard perspectives; INFO EXR, information experience; PR EXR, price experience; PT EXR, patient experience; ACC EXR, access experience; SERV EXR, services experience; BUILENV EXR, building environment experience; BUILCAP EXR, building capacity experience.

**Table 1 ijerph-19-07149-t001:** Sociodemographic and characteristics of the second panel (executive managers).

Sociodemographic Characteristic		Panelists *N*	%	Sociodemographic Characteristic		Panelists *N*	%
Age				Position			
	30–39 years	4	30.7		CMO	3	23.1
	40–49 years	7	53.8		CFO	3	23.1
	60–69 years	2	15.4		CEO	3	23.1
Gender					Managing director	3	23.1
	Male	7	53.8		Operation manager	1	7.7
	Female	6	46.2	Highest degree			
Academic background					Bachelor degree	8	61.5
	Medicine	4	30.8		Master’s degree	5	38.5
	Management	4	30.8	Administrative type			
	Accounting	3	23.1		Private	4	30.8
	Accounting and management	2	15.4		NGO	4	30.8
Years of experience					Public	5	38.5
	5–10 years	1	7.6				
	More than 10 years	12	92.3				

CMO, chief medical officer; CFO, chief financial officer; CEO, chief executive officer, NGO, nongovernmental organization.

**Table 2 ijerph-19-07149-t002:** Characteristics and sociodemographics of respondents (patients).

		Number of Patients (*N* = 740)	%			Number of Patients (*N* = 740)	%
Age (years)	Less than 20	63	8.5	Income (NIS)	Less than 1000	195	26.4
	20–29	209	28.2		1000–2000	98	13.2
	30–39	208	28.1		2001–3000	152	20.5
	40–49	159	21.5		3001–4000	140	18.9
	50–59	71	9.6		More than 4000	155	20.9
	60–69	24	3.2	Insurance type ^#^	Public	492	66.5
	More than 70	6	0.8				
Gender	Females	325	43.9		Private	143	19.3
					UNRWA	63	8.5
	Males	415	56.1		No insurance	109	14.7
Highest degree	Elementary	85	11.5	Number of the current visit	First	227	30.7
	Secondary	217	29.3		Second	187	25.3
	Bachelor	366	49.5		Third	91	12.3
	Masters	63	8.5		Fourth	54	7.3
	PhD	9	1.2		Fifth	181	24.5
Working sector	Public	175	23.6	Admission status	Inpatients	350	47.3
	Private	183	24.7		Outpatients	390	52.7
	Free lancer	156	21.1	Respondent opinion is based on ^#^	Personal experience	570	77
	Retired	17	2.3				
	Unemployed	209	28.2		Family experience	306	41.4
					Friends experience	96	13

NIS, New Israeli Shekel; UNRWA, The United Nations Relief and Works Agency for Palestine Refugees in the Near East; NGO, Non-Governmental Organization; ^#^, multiple response question.

**Table 3 ijerph-19-07149-t003:** Number of patients and hospitals based on hospital characteristics.

		Number of Patients (*N* = 740)	%	Number of Hospitals (*N* = 14)	%
Administrative Type	Public	252	34.1	5	36
	NGO	277	37.4	5	36
	private	159	21.5	3	21
	UNRWA	52	7	1	7
City	Hebron	150	20.3	3	21
	Jerusalem	86	11.6	1	7
	Nablus	249	33.6	5	36
	Qalqilya	52	7	1	7
	Ramallah	151	20.4	3	21
	Tulkarm	52	7	1	7
Area	North	353	47.7	7	50
	Middle	237	32	4	29
	South	150	20.3	3	21
Accredited hospital	Yes	185	25	3	21
	No	555	75	11	79
Size	Small (No. of beds <80)	241	32.6	5	36
	Medium (No. of beds 80–160)	261	35.3	5	36
	Large (No. of beds >160)	238	32.2	4	29

**Table 4 ijerph-19-07149-t004:** Exploratory factor analysis (EFA).

Component	Item	Item Code	Component/Item Loadings											
			1	2	3	4	5	6	7	8	9	10	11	12
BSCP ATT	I will recommend this hospital to my family and friends.	SAT3	0.894											
	I believe I receive an accurate medical examination at this hospital.	PIN1	0.783											
	I will choose this hospital again when I need a medical consultation.	PEN2	0.754											
	I believe this hospital offers me better treatment than the other Palestinian hospitals.	PEN3	0.686											
	My overall satisfaction with this hospital’s performance is high.	SAT1	0.683											
	I believe this hospital has a high cure rate.	PEN1	0.651											
	I will choose this hospital again when I need a medical consultation.	SAT2	0.579											0.556
	I believe the staff at this hospital are competent, knowledgeable, updated, and skilled.	PLE1	0.537											
PT EXR	This hospital distributes surveys to assess my satisfaction before discharge.	PCU4		0.968										
	This hospital distributes surveys to assess my needs upon arrival to the hospital, admission, or during the stay.	PCU3		0.755										
	Separate male/female waiting area are available at this hospital.	PEN9		0.655										
	This hospital follows up with me after the discharge.	PLE11		0.645										
	My complaints are taken seriously into consideration and solved immediately at this hospital.	PCU5		0.601										
	I can book an online or a phone appointment at this hospital easily.	PLE7		0.586										
	Staff trained me on infection precaution measures such as hand hygiene, cough etiquette, isolation rational, personal protective equipment, etc.	PLE6		0.560										
SERV EXR	Female doctors are available at this hospital.	PEN8			0.625									
	There are a variety of departments at this hospital.	PIN12			0.616									
	Services at night, vacations, and weekends are available at this hospital.	PIN18			0.556									
	There are a variety of specialties at this hospital.	PIN15			0.540									
PR EXR	I pay a reasonable price for the other medical services (laboratory, radiology, etc.) at this hospital.	PFI2				0.959								
	I pay a reasonable price for the medications at this hospital.	PFI3				0.888								
	I pay a reasonable price for the medical consultation at this hospital.	PFI1				0.848								
BUIL EXR	There is a sufficient number of chairs in the waiting area.	PEN13					0.639							
	The hospital has clean departments, corridors, rooms, bathrooms.	PEN12					0.585							
	The capacity of departments at this hospital including (ER, ICU, waiting room, etc.) is sufficient enough.	PEN14					0.562							
	This hospital has new building infrastructure (walls, ceiling, bathrooms, etc.).	PEN11					0.519							
ACC EXR	The accessibility to this hospital is easy by either public transportation or my car.	PEN4						0.910						
	The accessibility to this hospital in an emergency is easy.	PEN5						0.907						
COMP IMAGE	There is a probability for postoperative bacterial infection at this hospital	PIN6							0.765					
	There is a probability for case referral to another hospital	PIN5							0.752					
	There is a probability for case readmission at the same hospital	PIN4							0.602					
TECH IMAGE	This hospital use technology to link my prescriptions and tests with pharmacy and labs.	PLE9								0.842				
	This hospital use technology for saving my records.	PLE10								0.564				
INFO EXR	Information provided to me to be used after discharge is sufficient (medication and side effects, health condition, etc.).	PLE4									0.708			
HSRP IMAGE	I believe this hospital offers social and volunteering activities to the community.	PEN7										0.601		
	I believe this hospital offers exemptions for poor patients.	PEN6										0.566		
WT EXR	I wait for a long time before receiving the medical service at this hospital.	PIN9											0.556	
Percentage of Variance (%) Total variance = 63.29%			27.46	5.81	5.02	3.71	3.40	3.24	2.79	2.70	2.48	2.37	2.22	2.09
Eigenvalues			14.28	3.02	2.61	1.93	1.78	1.69	1.45	1.40	1.29	1.23	1.16	1.10

Note: BSCP ATT, patient attitude toward balanced scorecard perspectives; PT EXR, patient experience; SERV EXR, services experience; PR EXR, price experience; BUIL EXR, building experience; ACC EXR, access experience; COMP IMAGE, complications perceived image; TECH IMAGE, technology perceived image; INFO EXR, information experience; HSRP IMAGE, hospital social responsibility perceived image; WT EXR, waiting time experience.

**Table 5 ijerph-19-07149-t005:** Goodness-of-fit indices in EFA and CFA and results.

EFA [50,57]		CFA [70]	
Criteria for Good Fit [56,64]	Measurements	Criteria for Good Fit	Measurements
-KMO: 0.6: low adequacy 0.7: medium adequacy 0.8: high adequacy 0.9: very high adequacy -Bartlett’s test *p* value < 0.05 -Inclusion/exclusion criteria for the components: Eigenvalues ≥ 1 Visual assessment of Cattell’s scree plot. -Inclusion/exclusion criteria for the items: 1- The factor loading ≥ 0.50. 2- Factor loadings on the assigned construct ≥ all cross-loading of other constructs.	-KMO = 0.901 (Chi square = 9052.693, degrees of freedom = 1326) -Bartlett’s test *p* value < 0.001 -12 components which have Eigenvalues above 1 -Cumulative variance = 63.29%	- χ2/df < 5 and closer to zero - The *p* value > 0.05 - GFI - CFI - TLI - GFI, CFI, and TLI close to 0.95 - RMSEA < 0.06 - SRMR ≤ 0.08	χ2/df = 1.58 *p* value < 0.001 GFI = 0.901 CFI = 0.953 TLI = 0.944 RMSEA = 0.039 SRMR = 0.0439

Note: EFA, exploratory factor analysis; CFA, confirmatory factor analysis; KMO, Kaiser–Meyer–Olkin; χ2/df, minimum discrepancy divided by its degrees of freedom; GFI, goodness-of-fit index; CFI, comparative fit index; TLI, Tucker–Lewis’s Index; RMSEA, root mean square error of approximation; SRMR, standardized root mean square residual.

**Table 6 ijerph-19-07149-t006:** Constructs IIC, CTIC, and CR.

Construct	IIC (Min.–Max.)	CTIC (Min.–Max.)	CR	N of Items (Total = 34)
COMP IMAGE	0.395–0.411	0.474–0.486	0.664	3
TECH IMAGE	0.390–0.594	0.486–0.642	0.794	3
BSCP ATT	0.328–0.641	0.505–0.735	0.861	9
INFO EXR	0.389–0.531	0.501–0.609	0.750	3
PR EXR	0.509–0.725 ^>>^	0.596–0.760 ^>>^	0.948	3
PT EXR	0.413–0.678	0.552–0.736	0.841	5
ACC EXR	0.853	0.853	0.906	2
SERV EXR	0.360	0.360	0.502	2
BUILENV EXR	0.412	0.412	0.643	2
BUILCAP EXR	0.527	0.527	0.721	2

COMP IMAGE, complications perceived image; TECH IMAGE, technology perceived image; BSCP ATT, patient attitude toward balanced scorecard perspectives; INFO EXR, information experience; PR EXR, price experience; PT EXR, patient experience; ACC EXR, access experience; SERV EXR, services experience; BUILENV EXR, building environment experience; BUILCAP EXR, building capacity experience; IIC, inter-item correlation; CITC, corrected item total correlation; CR, composite reliability; ^>>^, was calculated only for patients who pay at the evaluated hospitals.

**Table 7 ijerph-19-07149-t007:** Convergent, discriminant, and divergent validity for the independent constructs.

Construct	AVE	INFO EXR	PR EXR	PT EXR	ACC EXR	SERV EXR	BUILENV EXR	BUILCAP EXR
INFO EXR	0.501	**0.708**						
PR EXR	0.858	*0.084 **	**0.926**					
PT EXR	0.515	*0.507 ***	*0.095 **	**0.718**				
ACC EXR	0.828	*0.121 ***	*-0.005*	*0.053*	**0.910**			
SERV EXR	0.337	*0.341 ***	*0.002*	*0.242 ***	*0.164 ***	**0.581**		
BUILENV EXR	0.477	*0.302 ***	*-0.006*	*0.336 ***	*0.110 ***	*0.209 ***	**0.691**	
BUILCAP EXR	0.564	*0.288 ***	*0.016*	*0.366 ***	*0.164 ***	*0.238 ***	*0.394 ***	**0.751**

Note: PT EXR, patient experience; INFO EXR, information experience; PR EXR, price experience; COMM EXR, communication experience; ACC EXR, access experience; BUILCAP EXR, building capacity experience; TECH EXR, technology experience; DEPV EXR, departments variety experience, SERV EXR, services; WT EXR, waiting time experience; BUILENV EXR, building environment experience; AVE, average variance extracted calculated by the average square of loadings at each construct and used to evaluate the convergent validity; **Bold**, square roots of the average variance extracted; *Italic*, Spearman correlations between independent constructs. Both are used to evaluate discriminant validity; *, *p* < 0.05; **, *p* < 0.01.

**Table 8 ijerph-19-07149-t008:** Convergent, discriminant, and divergent validity for the dependent constructs.

Construct	AVE	BSCP ATT	TECH IMAGE	COMP IMAGE
BSCP ATT	0.413	**0.643**		
TECH IMAGE	0.564	*0.397 ***	**0.751**	
COMP IMAGE	0.400	*0.216 ***	*0.156 ***	**0.633**

COMP IMAGE, complications perceived image; TECH IMAGE, technology perceived image; BSCP ATT, patient attitude toward balanced scorecard perspectives; AVE, average variance extracted calculated by the average square of loadings at each construct and used to evaluate the convergent validity; **Bold**, square roots of the average variance extracted; *Italic*, Spearman correlations between independent constructs, both are used to evaluate discriminant validity; **, *p* < 0.01.

## Data Availability

The datasets generated and/or analyzed during the current study are not publicly available because the data are still not fully analyzed and the research is still in process but available from the corresponding author (F.A.) upon reasonable request, with the permission of the UNRWA, Palestinian Ministry of Health, and Al Makassed Hospital.

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
