# Peer review of "Assessing Patient Experience and Attitude: BSC-PATIENT Development, Translation, and Psychometric Evaluation—A Cross-Sectional Study"

_ijerph, 2022, doi:10.3390/ijerph19127149_

Round 1
Reviewer 1 Report
Dear Authors
Thank you for the opportunity to read your work.I would like to congratulate you on a well designed study and good written paper.The paper is written clearly and takes almost all factors into cosideration.The contribution is innovative.
Please see minor revisions.

Author Response
All the suggested minor revisions were addressed.
We want to thank the reviewer for their time and evaluation. Finally, if the reviewer still see that any amendments are essential or needed, the authors are willing to perform them as soon as possible.

Reviewer 2 Report
This manuscript provides the development of a questionnaire for patient engagement in balanced score card implementations, which offers an instrument for evaluation of patient experiences.
The authors indicate the contextual issues contributing to the need for the research (line 30), with an explanatory defining of the conceptual model and primary components of the BSC-Patient (line 76). However, it may be of value here to expand the review of literature to incorporate a broader understanding and perspective of the BSC approach. Can you expand a bit on the history of these methods? Why were these particular components selected to represent BSC-Patient? Have other similar methods been tried? Were they successful? What are the current best practices? While this section of the manuscript does offer a well-developed conceptual framework, it would be beneficial to provide a stronger contextualization of your proposed approach.
In the discussion (beginning line 471), I would likewise suggest a more developed assessment of the research findings in the context of the existing literature. In general, a desired outcome of research is to compare and contrast the results with extant knowledge, to determine the contribution of the research and to contextualize our current understanding of the topic. While, again, the authors do provide a good interpretation of the results within existing literature, an expansion of this section would add strength to the manuscript. Likewise, it would be helpful to have presented some of this literature earlier in the manuscript (conceptual framework) to offer the reader a better comprehension of current BSC studies and other validated instruments.
Somewhere in relation to the ‘Strengths and Limitations’ (line 552) and the ‘Conclusion’ (line 585) of the manuscript, there should be a broader discussion of implications. In other words, given the results of your research, taking into consideration the strengths and limitations, what is the impact of this research? What are the implications for practice? For future research? What does this mean for our understanding of BSC instruments? What is the overall contribution to the existing literature? To the field? Again, a more expansive discussion here of implications will help to strengthen the overall manuscript.
Author Response
First, we want to thank the reviewer for their time in considering this manuscript for review and the suggested modifications to improve it.
All the suggested modifications were made. Moreover, we run the manuscript in AJE to check the English language and the amendments were performed accordingly.
Finally, if the reviewer still sees that any amendments are essential or needed, the authors are willing to perform them as soon as possible.

Reviewer 3 Report
Comments to the paper ID: ijerph-171536-6
Assessing patient experience and attitude: BSC-PATIENT development, translation, and psychometric evaluation: a cross sectional study
General comments
The authors propose the creation and validation of an instrument to integrate the experiences and attitudes of hospital patients in Palestine into the "balanced score card (BSC)". The main contribution is the development of a structured assessment instrument to capture patients' perceptions and experiences in relation to the topics or areas of interest of this model.
There is a gap regarding the background and relevance that this model (Balance Score Card) has in the field of performance evaluation in general, and in health management in particular. It was necessary to consult other sources to understand their origins and possible relevance. The model is adapted from the field of administration and marketing, but it is necessary to argue why it would be relevant to the field of health management, why that model and not others, or what are its main advantages.
Timely comments
In the summary and introduction, it is necessary to describe what it is and what are the origins of the Balance Score Card, why it can be relevant for health management, and why it would require integrating the vision of patients. Although a "Conceptual Framework" section is included in relation to the Balance Score Card, it is not clear which health model that covers or integrates this proposal. It is necessary to clarify why we must "bet" on this model of performance evaluation and not another, or why it is relevant to bring it from one field of knowledge (marketing-finance) to another. A review of the way in which these issues have been evaluated in health or existing instruments is also required.
The method in general adequately describes the process of creating the instrument. The process of "translation, retranslation" is somewhat unnecessary, this would only make sense if you want to validate the instrument also in the English-speaking population, but the argument of "the dimensions found were in English" is not enough. Furthermore, was the creation of the first items in English done by Experts in the English language?
Regarding the calculation of the sample size, there are other criteria for its selection, especially in the field of psychometrics, where priority is given for example in number of items or response categories to calculate the sample.
The proposed statistical analysis is adequate given the objective of the work. The tables in the results section require edits to facilitate their visualization, for example, it is unnecessary to place the reagents that did not load in any factor, especially if their value is not reported. The confirmatory Factor Analysis model also requires editing.
The discussion section can be improved, starting by highlighting that it was possible to build a valid and reliable instrument, its usefulness compared to other instruments in the field of health, and its contribution in that area. This section makes a comparison of the disadvantages of other instruments (again that have been developed from marketing), but does not compare them with health evaluations, in addition it may be more appropriate to include them in the "Introduction" section.
Author Response
Thank you so much for your time and the deep analysis of this paper. We performed all the suggested modifications to improve this manuscript, as described below. Also, we run the manuscript in the AJE and the amendments were performed to improve the English language.
Finally, if the reviewer still sees that any amendments are essential or needed, the authors are willing to perform them as soon as possible.

Round 2
Reviewer 3 Report
I would like to thank the authors for their effort in improving this article.
All the suggested modifications were made.
Many thanks!
